# Computed Tomography-Based Quantitative Texture Analysis and Gut Microbial Community Signatures Predict Survival in Non-Small Cell Lung Cancer

**DOI:** 10.3390/cancers15205091

**Published:** 2023-10-21

**Authors:** David Dora, Glen J. Weiss, Zsolt Megyesfalvi, Gabriella Gállfy, Edit Dulka, Anna Kerpel-Fronius, Judit Berta, Judit Moldvay, Balazs Dome, Zoltan Lohinai

**Affiliations:** 1Department of Anatomy, Histology and Embryology, Semmelweis University, 1094 Budapest, Hungary; dora.david@med.semmelweis-univ.hu; 2Department of Medicine, UMass Chan Medical School, Worcester, MA 01655, USA; glen.weiss@umassmed.edu; 3Department of Tumor Biology, National Koranyi Institute of Pulmonology, 1122 Budapest, Hungary; megyesfalvi.zsolt@semmelweis-univ.hu (Z.M.); berta.judit@koranyi.hu (J.B.); moldvay.judit@koranyi.hu (J.M.); 4Department of Thoracic Surgery, National Institute of Oncology, Semmelweis University, 1122 Budapest, Hungary; 5Department of Thoracic Surgery, Comprehensive Cancer Center, Medical University of Vienna, 1090 Vienna, Austria; 6Pulmonary Hospital Torokbalint, 2045 Torokbalint, Hungary; galffy.gabriella@torokbalintkorhaz.hu (G.G.); dulka.edit@torokbalintkorhaz.hu (E.D.); 7Department of Radiology, National Koranyi Institute of Pulmonology, 1122 Budapest, Hungary; 8Department of Translational Medicine, Lund University, 22184 Lund, Sweden; 9Translational Medicine Institute, Semmelweis University, 1094 Budapest, Hungary

**Keywords:** computed tomography-based texture analysis, artificial intelligence, advanced NSCLC, PD-L1, microbiome

## Abstract

**Simple Summary:**

There is a lack of understanding of the pathogenesis and mechanisms accounting for the large variability in tumor response to immune checkpoint inhibition. In this study, we investigate the role and composition of the human gut microbiome in the clinical setting by integrating shotgun metagenomics and quantitative texture analysis (QTA) of CT images in NSCLC patients treated with anti-PD-L1 immunotherapy using a novel machine learning approach. Using all available parameters, the XGB machine learning system predicted therapeutic response with an accuracy of 83% and correctly separated long-term survival patients from short-term survival patients with an accuracy of 69%. Our findings show that an integrated signature of these characteristics may predict outcomes more accurately than separate measures and may have potential therapeutic implications in the future.

**Abstract:**

This study aims to combine computed tomography (CT)-based texture analysis (QTA) and a microbiome-based biomarker signature to predict the overall survival (OS) of immune checkpoint inhibitor (ICI)-treated non-small cell lung cancer (NSCLC) patients by analyzing their CT scans (*n* = 129) and fecal microbiome (*n* = 58). One hundred and five continuous CT parameters were obtained, where principal component analysis (PCA) identified seven major components that explained 80% of the data variation. Shotgun metagenomics (MG) and ITS analysis were performed to reveal the abundance of bacterial and fungal species. The relative abundance of Bacteroides dorei and Parabacteroides distasonis was associated with long OS (>6 mo), whereas the bacteria Clostridium perfringens and Enterococcus faecium and the fungal taxa Cortinarius davemallochii, Helotiales, Chaetosphaeriales, and Tremellomycetes were associated with short OS (≤6 mo). Hymenoscyphus immutabilis and Clavulinopsis fusiformis were more abundant in patients with high (≥50%) PD-L1-expressing tumors, whereas Thelephoraceae and Lachnospiraceae bacterium were enriched in patients with ICI-related toxicities. An artificial intelligence (AI) approach based on extreme gradient boosting evaluated the associations between the outcomes and various clinicopathological parameters. AI identified MG signatures for patients with a favorable ICI response and high PD-L1 expression, with 84% and 79% accuracy, respectively. The combination of QTA parameters and MG had a positive predictive value of 90% for both therapeutic response and OS. According to our hypothesis, the QTA parameters and gut microbiome signatures can predict OS, the response to therapy, the PD-L1 expression, and toxicity in NSCLC patients treated with ICI, and a machine learning approach can combine these variables to create a reliable predictive model, as we suggest in this research.

## 1. Introduction

Immune checkpoint inhibitors (ICI) are frequently administered as first-line therapy in non-small cell lung cancer (NSCLC). Notably, however, only 15–20% of the ICI-treated tumors show stable disease (SD) or response to therapy, and only a portion of the patients experience durable benefit [1,2,3]. To date, there are no other widely used reliable biomarkers for ICI treatment eligibility in routine clinical practice. Understanding and non-invasively assessing the host and tumor microenvironment might improve both therapeutic and survival outcomes [4].

Computed tomography (CT)-based quantitative texture analysis (QTA) represents a non-invasive diagnostic method in different cancer types; it converts digital images into high-dimensional data, enabling the quantification of spectral properties, gray-level patterns, and pixel interrelationships [5,6]. The CT image of the primary tumor can reveal heterogeneity in the density that might be associated with distinct characteristics of the tumor microenvironment (TME) or tumor-infiltrating immune cells [7]. T-cells represent a hallmark of ongoing immune surveillance with potential therapeutic importance. Increased tumor immune cell infiltration predicts survival in lung cancer [8]. Even though T-cell infiltration of solid tumors is associated with favorable patient outcomes, the mechanisms underlying variable immune responses between individuals are not well understood [9,10,11]. Therefore, QTA parameters might represent a prognostic and predictive biomarker for ICI-treated patients.

Recent data report an association between the gut microbiome and ICI efficacy [12]. The immunology of the gut–lung axis is an emerging field and can be explained by antigen mimicry or cross-reactivity [13]. Microbiota antigens that pass the intestinal barrier can result in T-cell priming, stimulating cytokine and interferon production and eliciting an anti-tumor immune response. Therefore, gut bacteria might regulate tumor-infiltrating immune cells. Fungal species in the gut interact with bacterial growth and may also be associated with lung diseases. Certain fungal species can direct immune cell trafficking, especially the inflammatory or tolerant immune responses that can emerge or evolve [14,15,16]. Cancer and chronic obstructive pulmonary disease (COPD) have recently been linked to dysbiotic airway microbiota and commonly occur alongside gastrointestinal (GI) disorders [17]. Others report a key role for Bacteroidales in the immunostimulatory effects of the ICI blockade [18,19]. Recent studies revealed a direct linkage between the gut microbiome’s composition and ICI efficacy in malignant melanoma and NSCLC [20,21,22,23]. 

There is a lack of understanding of the pathogenesis and mechanisms accounting for the large variability in tumor response to ICI. However, the clinical significance and potential theranostic role of QTA and the gut mycobiome in lung cancer have not yet been explored. Here, we investigate the role and composition of the human gut microbiome in the clinical setting by integrating shotgun metagenomics (MG) and QTA in NSCLC patients treated with anti-PD-L1 immunotherapy, using the novel XBoost machine learning (ML) approach. Furthermore, we aimed to explore whether an ML-integrated signature obtained from radiomics and metagenomics could provide a more reliable prognostic and predictive nomogram than the analysis of these features separately.

## 2. Materials and Methods

### 2.1. Ethical Statement

This current work was conducted in accordance with the Helsinki Declaration of the World Medical Association study guidelines. The national ethics committee (Hungarian Scientific and Research Ethics Committee of the Medical Research Council (ETTTUKEB-50302-2/2017/EKU)) officially approved the study. All the patients involved/recruited consented to the study. After the clinicopathological data were collected, the patient identifiers were removed; so, the patients cannot be identified directly or indirectly.

### 2.2. Study Population and Treatments

A total of 129 advanced stage NSCLC patients treated with ICI were enrolled in our study; they received standard-of-care nivolumab or pembrolizumab treatments (46% received first-line and 54% second-line ICI) between 2017 and 2018 at the National Koranyi Institute of Pulmonology, Budapest, Hungary, and at the County Hospital of Pulmonology, Torokbalint, Hungary. All the patients included were diagnosed with advanced-stage disease (Stage IIIB/IV). We included patients with histologically confirmed adenocarcinoma (ADC), squamous cell carcinoma, and non-small cell lung carcinoma not otherwise specified (NSCLC-NOS). The clinical TNM (tumor, node, metastasis) stage according to the Union for International Cancer Control (8th edition) and age at the time of diagnosis were recorded. CT scans of all 129 patients and stool samples from 58 of these patients were available and analyzed. The clinicopathological data included gender, age, stage, and PD-L1 tumor proportion score (TPS); the COPD stages according to the Global Initiative for Chronic Obstructive Lung Disease (GOLD) criteria; forced expiratory volume in 1 s (FEV1); medications, including proton pump inhibitors (PPIs), steroids, antibiotics, antifungals, and supplements before and after treatment; response to therapy; and overall survival (OS). The patients were classified according to PD-L1 TPS with a cutoff of 50% (high vs. low) and a cutoff of 1% (positive vs. negative). Immunotherapeutic agents were administered as first-line therapy if the PD-L1 tumor proportion score (TPS) was ≥50% and second-line therapy if the PD-L1 TPS was <50%. Before second-line immunotherapy, the patients received standard-of-care platinum-based chemotherapy. Responders were distinguished from non-responders according to the RECIST criteria, where patients with complete response (CR), partial response (PR), or stable disease (SD) were assessed as responders and patients with progressive disease (PD) were assessed as non-responders after 12 weeks of ICI treatment. OS was calculated from the time of diagnosis to death or the last available follow-up. Smoking was scored from 1 to 4 (never, passive, former, current) and defined in pack years (PY). Any type of immune-related adverse event (irAE) reported after ICI treatment was graded based on the ESMO clinical practice guidelines, and the data were analyzed according to the presence of any type of irAE vs. no-irAE reported. Each irAE was characterized either as a binary variable (toxicity—no toxicity) or as a continuous variable (grade). COPD severity was also determined according to the COPD Assessment Test (CAT) score. The date of the last follow-up included in this analysis was February 2021. The therapeutic approaches across all the centers were conducted in line with the current National Comprehensive Cancer Network guidelines. The patients were classified based on their OS into short-term (≤6 months) versus long-term (>6 months) survivors.

### 2.3. QTA and Principal Component Analysis

QTA was applied for pretreatment CT images of primary NSCLC tumors in the lung (*n* = 129). 

Based on a standard-of-care standardized radiology report using RECIST 1.1 criteria in the reporting, a board-certified radiologist had a second look and checked the tumor region of interest. The radiologist selected the primary tumor, the most extensive diameter lesion surrounded by lung parenchyma and not centrally located, directly infiltrating, or connected with other tissue compartments in the mediastinal area or chest wall. CT scans were excluded that did not meet these criteria. Three-dimensional tumor segmentation was performed using the 4.10 version of the 3D Slicer, and a total of 105 CT parameters from each CT image were obtained. The 3D segmentation was performed using an automated (Fast GrowCut), which is an in-built robust algorithm to segment the volume fully. We used Label 1 as the foreground (tumor) and Label 2 as the background (lung parenchyma). Multiple label colors can be used to define regions representing parts of anatomical structures. The algorithm finds the best labeling for an adjacent pixel to match the tumor volume. Next, we used the Pyradiomics software package (https://pyradiomics.readthedocs.io/en/latest/features.html, accessed on 1 March 2022) to obtain the QTA parameters from the segmentation masks and the CT scans. 

We used the Sklearn machine learning library in Python for data preprocessing and standardization, reducing the number of CT parameters by principal component analysis (PCA), which identified seven primary components (PCs) that explained 80% of the data variation. The components thus obtained were further analyzed with hierarchical cluster analysis. Long-term versus short-term survivors, responders versus non-responders, and patients with PD-L1 expression (<50% vs. ≥50%) were analyzed based on naïve Bayes and k-means clustering, and the seven principal components (PCs) were incorporated into the machine learning models. We verified the accurateness of the machine learning algorithms with leave-one-out cross-validation.

### 2.4. PD-L1 Immunohistochemistry and TPS Scoring

For PD-L1 immunohistochemistry (IHC) analysis, the tumor samples obtained through lung biopsy were accessible for *n* = 125 advanced-stage NSCLC patients. To perform the IHC staining, 4 µm sections were cut from formalin-fixed paraffin-embedded (FFPE) blocks. Utilizing a rabbit monoclonal antibody for PD-L1 diluted at 1:300 (CST, cat: 13684S), staining was executed on a Leica Bond RX autostainer. A Bond Polymer Refine Detection kit (#DS9800) was used and followed Leica IHC Protocol F, while epitope retrieval was performed for twenty minutes under low pH conditions. Afterwards, the slides were subjected to clearing and dehydration on a Tissue-Tek Prisma platform before being coverslipped with Tissue-Tek Film. Next, a proficient histopathologist evaluated the PD-L1 expression using the FDA-approved TPS scoring system. All *n* = 125 patients received a positive (TPS ≥ 1%) or negative (TPS < 1%) classification and *n* = 71 patients were assessed and categorized as PD-L1-high (TPS ≥ 50%) or -low (TPS < 50% percentile) based on further scoring.

### 2.5. DNA Extraction from Stool Samples

Baseline stool samples were obtained simultaneously within seven days, before or after the first cycle of ICI administration and after signed informed consent was obtained. All the samples were placed on the day of collection in the −80 °C freezer until the isolation and sequencing steps. The stool samples were processed according to Novogene protocol, as previously described [24]. In brief, after thorough mixing with CTAB lysis buffer, the samples were incubated at 65 °C and centrifuged at 12,000× *g* for 5 min at 4 °C. Nine hundred microliters of phenol:chloroform:isoamyl alcohol (25:24:1, pH = 6.7; Sigma-Aldrich, Taufkirchen, Germany) was added to the supernatants and mixed thoroughly prior to incubation at room temperature for 10 min. After phase separation, the samples were centrifuged at 12,000× *g* for 10 min at 4 °C. DNA precipitation was obtained by adding 450 μL of isopropanol (Sigma-Aldrich) containing 50 μL of 7.5 M ammonium acetate (Thermo Fisher, Waltham, MA, USA) to the upper phase from the last extraction step. After being washed in 70% ethanol, the DNA pellets were air-dried and re-suspended in 200 μL of 75 mM TE buffer (pH = 8.0; Sigma-Aldrich). 

### 2.6. Library Preparation and MG Sequencing

The generation of the sequencing library was based on Illumina technologies, following the manufacturers’ recommendations. Briefly, genomic DNA was randomly fragmented to a size of 350 bp, and the fragments were carefully size-selected with sample purification beads. Next, the selected fragments were A-tailed, end-polished, and ligated with an adapter. After one more sequence of bead purification, the fragments were amplified by PCR reaction. After analysis for size distribution by real-time PCR, the library was sequenced on an Illumina platform Novaseq 6000 (Novogene, Beijing, China) with paired-end reads of 150 bp.

### 2.7. Internal Transcribed Spacer (ITS2) Sequencing

For ITS2 sequencing, genomic DNA concentration was determined by Qubit. Two hundred nanograms of DNA was used as an input for the PCR reaction with the corresponding primer set, specifically binding to different hypervariable regions, where a unique barcode was assessed for each primer. The purified PCR product was then utilized as a template to create a library. The PCR products were pooled together in equal proportions before they were A-tailed, end-polished, and adapter-ligated. The library was analyzed for size distribution and quantified using real-time PCR after bead filtering and PCR amplification (to make the library entirely double-stranded). As previously disclosed [24], library sequencing was carried out on a Hiseq2500 platform.

### 2.8. Quality Control

The Sunbeam 2.1 pipeline was used to perform the quality control of the raw reads, as previously described [25]. Briefly, „cutadapt” version 2.8 was used to remove the universal adapter sequences, and „trimmomatic” version 0.36 [26] was used to perform Illumina-specific adapter trimming, window quality trimming (Q5 over 25 nt), and 3’ and 5’ clipping (Q < 6). Reads shorter than 36 nt were removed. To remove contamination from the host-derived human reads, BWA version 0.7.17 was used [27] against a masked human reference genome (GRCh38-89). Reads with 99% coverage and >97% identity with the human reference were removed. 

### 2.9. Microbial Taxonomic Profiling

High-quality reads were taxonomically annotated using MetaPhlAn2 (version 2.7.7) with default parameters to determine relative abundances of bacterial species. The PIPITS pipeline (version 2.4) with default parameters was utilized for taxonomic annotation of fungal ITS, as previously described [24,28]. The remaining reads were binned using the mothur classifier and aligned to the UNITE fungus database based on 97 percent similarity as operational taxonomic units [29].

The differentially abundant taxa were identified using the Wald test implemented in the R package DESeq2 v1.22.2 on the unrarefied relative abundance data. The statistical significance was filtered with FDR-corrected *p* <0.05, unless otherwise stated. Logarithmic normalization was used to overcome extensive variations in the MG expression data. We conducted a Student’s *t*-test to analyze the difference between the log-normalized expression distributions in each species’ long OS and short OS groups. Among the patients with MG data available, 44 long OS and 14 short OS groups were identified. For each species, we conducted a Student’s *t*-test to find out whether the difference between the distributions of the log-normalized expressions in the long OS and short OS groups were significant with *p* = 0.01. Amongst the 901 species, we found significant differences in 22, and the log-normalized expressions of these species were incorporated into the machine learning models. Due to multiple testing, we adjusted the Benjamini–Hochberg false discovery rate correction; with FDR = 0.25, we found that the 6 most substantial results from the 22 selected species were still significant. For these species, we calculated and plotted the log fold change between the groups.

### 2.10. XGBoost Models for Classification

XGBoost, or extreme gradient boosting, is a decision tree-based gradient boosting algorithm used for our machine learning approach to generate predictive values for clinical outcomes. Many examples show that XGBoost can outperform other decision tree-based learning algorithms (https://github.com/dmlc/xgboost/tree/master/demo#machine-learning-challenge-winning-solutions, accessed on 1 May 2022). The XGBoost is similar to the random forest algorithm, which is an ensemble version of the decision tree (https://www.javatpoint.com/machine-learning-decision-tree-classification-algorithm, accessed on 1 May 2022). The prediction of a random forest algorithm is based on the sum of the individual decision tree predictions. Gradient boosting means that the model not only selects a random set of weights for each of the decision trees but trains them in a sequence: after the first tree is created, it calculates the error for the predictions with a loss function. After that, each new tree trains to predict the errors of the previous one by assigning higher weights or importance for the mislabeled entries. XGBoost is a C++-based implementation of gradient boosting, with additional weight regularization to prevent overfitting and parallelization for multiple CPU cores.

The regressor version of the XGBoost algorithm was used to handle the categorical and continuous data (relative occurrence of bacterial and fungal species in patients and QTA signatures). This also provided numerical prediction values, which we then cast into the numerical categories of the predicted property. Training and prediction were also performed using the incomplete dataset. Filling the incomplete dataset was conducted by the XGBoost regressor algorithm by predicting the missing values for a property from a model trained on the subset of the incomplete dataset. In the first experiment, we used all the available features of the data to create a single model, and we trained it with the missing values. Then, in the second experiment, we applied an iterative method, where we started with the patients with all the available features; then, we predicted the missing features with different XGBoost models. This method is prone to overfit because it uses the predictions of predictions for training; so, we tested it with multiple pseudo-random training-validation splits, but on average, it achieved better results than the first experiment.

## 3. Results

A total of 129 NSCLC patients with CT scans were included in our analysis. MG data were available for 44 long-term and short-term survivors. The study design, patient cohorts, and availability of clinical and treatment data are shown in Figure 1. A summary of the clinical data of the patients is shown in Table 1.

### 3.1. QTA Parameters Can Predict OS, Response to Therapy and PD-L1 Expression

First, we performed principal component analysis (PC) on the 105 QTA features and visualized patient groups (OS, response, and PD-L1 expression) according to PC1 and PC2 (Figure 2A–C). Next, we assessed individual QTA parameters to evaluate their association with OS, the response to therapy, and PD-L1 expression. Between the long- and short-term survivors, five QTA parameters showed significant differences. Coarseness was increased in patients with long OS (*p* < 0.001, Figure 2D), whereas Energy (*p* = 0.04, Figure 2E), Kurtosis (*p* = 0.048, Figure 2F), and Surface Area (*p* = 0.019, Figure 2G) were increased in individuals with short OS. Regarding the therapeutic response, parameters such as ClusterTendency (*p* = 0.019, Figure 2H), Complexity (*p* = 0.049, Figure 2I), and Variance (*p* = 0.045, Figure 2J) showed significantly increased values in responder patients. Interestingly, the parameters Coarseness (*p* = 0.006, Figure 2K), Energy (*p* = 0.011, Figure 2M) and Kurtosis (*p* = 0.038, Figure 2O) also showed a significant difference in terms of PD-L1 expression (50%) (similar to that of OS). In addition, Contrast (*p* = 0.033, Figure 2L) was significantly higher in PD-L1-high patients, whereas GrayLevelVariance (*p* = 0.044, Figure 2N) was significantly increased in PD-L1-low (<50%) patients. 

### 3.2. Microbial Taxonomic Profiling Reveals Associations with OS, Response to Therapy, PD-L1 Expression, and Toxicity

Next, we analyzed the associations of microbial species, principal components, clinicopathological parameters, and drugs administered before or after the first ICI cycle, as described in the methods. Figure 3A shows microbial taxa that significantly correlate with OS. Interestingly, Bacteroides dorei and Parabacteroides distasonis were negatively correlated with other significant species included in our analysis. Also, Bacteroides dorei and Parabacteroides distasonis were more abundant in patients with long OS (vs. short-term survivors) (Figure 3A, Table 1). In contrast, the bacterial species Clostridium perfringens and Enterococcus faecium; the fungal species Cortinarius davemallochii, Helotiales, and Chaetosphaeriales; and the fungal class Tremellomycetes showed significantly increased abundance in patients with short OS compared to those with long OS (Figure 3A, Appendix A) In patients with high-PD-L1-expressing tumors, Dorea formicigenerans and Lachnospiraceae bacterium showed significantly decreased relative abundances, whereas bacteria Enterococcus avium and Streptococcus tigurinus and fungi Hymenoscyphus immutabilis and Clavulinopsis fusiformis showed significantly increased relative abundances (Figure 3B, Appendix A). The Bacterial species Lachnospiraceae and the fungal family Thelephoraceae were the most abundant in patients with treatment-related toxicities, while the fungal taxa Cutaneotrichosporon cutaneum and Rozellomycota were the most abundant in the AE-free patients (Figure 3C, Appendix A).

### 3.3. Correlation of Clinicopathological Parameters with Metagenome and Principal Components of QTA Analysis

Figure 4A shows the correlation between the clinicopathological parameters, OS, and microbial taxa, along with the *p*-values indicated for each association (Figure 4B). There was a negative correlation between the patients with COPD and those with high PD-L1 (≥50%) expression (r = 0.26; *p* = 0.029). FEV1% showed a significant negative correlation with the Clostridium celatum (r = 0.54; *p* = 0.006), Cortinarius davemallochii (r = 0.57; *p* = 0.004), Thelephoraceae (r = 0.44; *p* = 0.034), and Helotiales species (r = 0.43; *p* = 0.036). There were no significant associations between smoking and species that were associated with OS, but there was a significant positive correlation between smoking and treatment-related toxicity (r = 0.39; *p* = 0.003).

Further analysis revealed a negative correlation between steroid use and any toxicity grade (r = 0.27; *p* = 0.041). Pseudoprogression, an initial increase in tumor size followed by a decrease in tumor burden due to reactive immune cell infiltration following ICI administration [30], was associated with body mass index (BMI) (r = 0.33; *p* = 0.005) and the line of ICI (r = 0.31; *p* < 0.007). Interestingly, BMI was also associated with Enterococcus faecium, which showed a significant negative correlation with the parameter (r = 0.32; *p* = 0.018) that was confirmed by other studies [31,32]. The fungal taxa Serendipitaceae (r = 0.37; *p* = 0.046) and Hyphodiscus (r = 0.41; *p* = 0.022) showed significant association with proton pump inhibitor (PPI) or H blocker use before or after therapy initiation at any time point. The antibiotics administered both before and during therapy showed no significant correlation with response to ICI treatment. Appendix A shows the correlation coefficients (r) and *p*-values for all the parameters.

To create a compressed representation of the QTA parameters, PCA was performed, where seven PCs were identified. The seven components are linear combinations of the original CT features. The 105 QTA parameters and their contribution to the seven PCs are shown in Appendix A. The visual representation of the PCs according to the QTA parameters is shown in Appendix A. PC5 showed association with pseudoprogression (r = 0.39; *p* = 0.004), gender (r = 0.2; *p* = 0.021), and the presence of COPD (r = 0.28; *p* = 0.045). PC7 showed association with response to therapy (r = 0.423; *p* = 0.0015) and BMI (r = 0.31; *p* = 0.026). Among the microbial taxa, Clostridium perfrigiens was associated with PC1 (r = 0.32; *p* = 0.047) and the fungal order Helotiales showed significant correlation with PC3 (r = 0.35; *p* = 0.026).

### 3.4. Outcomes Predicted by the XGB Machine Learning Algorithm

The machine learning algorithm extreme gradient boosting (XGB) identified, with 84% and 81% accuracy, an MG signature for patients with a favorable response to ICI and long OS, and it had 79% accuracy in the prediction of high PD-L1 expression (≥50%). The QTA signature had 72% accuracy in predicting OS and 62% accuracy in predicting high PD-L1 (≥50%) expression (Figure 5A).

Next, we analyzed the role of the drugs and the outcomes, including PPIs and H blockers, steroids, antibiotics, antifungals, and supplements. Due to case numbers, we could not analyze the role of any of the drugs individually; instead, we pooled the drugs together in the machine learning algorithm, as mentioned above. Accordingly, XGB could predict OS with 61% accuracy, response to therapy with 71% accuracy, and toxicity with 70% accuracy from the data on the drugs prescribed before or during ICI administration. (Figure 5A). Consequently, we did not see a clinically relevant difference in the drug-related effect of timing. Figure 5B shows the negative predictive values (NPVs), while Figure 5C shows the positive predictive values (PPVs) generated by the XGB algorithm for different parameters separately and in combination. We included other clinicopathological parameters, such as age, gender, smoking (1–4), smoking pack year (PY), CAT score, FEV1, COPD, and line of ICI (Figure 5A–C, and others). The XGB algorithm was able to predict PD-L1 (≥50%) expression and response to ICI from these parameters with 83% and 76% accuracy.

The PPVs for the responses to ICI therapy derived from the QTA and MG signatures were 76% and 87%, respectively. In contrast, the NPVs for the same parameters reached only 29% and 58%, respectively (Figure 5B). The PPVs for OS from the QTA and MG signatures were 81% and 86%, respectively (Figure 5C). In contrast, The NPV for OS from MG was 62%; when combined with QTA, it was 45%, and when combined with the drugs, it was 65% (Figure 5B). Of note, the NPV of QTA and MG was the highest for PD-L1 (≥50%, 80%) and pseudoprogression (79%). The highest PPV was measured for MG and QTA to predict OS (86% and 81%, Figure 5C). Additionally, when combining MG and QTA, the PPV for the response to ICI therapy and OS was 90%.

The combination of three or more parameters further increased the PPVs for survival. MG, QTA, and the drug PPVs for the response to ICI therapy reached 91% and 86% for OS (Figure 5C). Combining all the parameters (MG, QTA, the drugs, and others) gave a PPV of 86% for the response to ICI therapy and 72% for OS. In contrast, the NPVs for OS in the latter three- or fourfold combinations only reached 45% and 31%, but for PD-L1 (≥50%, 80%, and 93%, respectively) and pseudoprogression (74% and 69%, respectively) it reached higher values (Figure 5C). The accuracy, PPVs, and NPVs not filled with the XGB algorithm are shown in Appendix A.

## 4. Discussion

The TME and immune cell infiltration in primary and metastatic tumors are key factors for ICI efficacy. Non-invasive assessment of the TME is an emerging field and may help in selecting patients for immunotherapy [33]. QTA can be used to quantitatively analyze the texture and heterogeneity of primary tumors [34,35] and lymph nodes [36]. To our knowledge, this is the first study investigating the associations of ICI outcomes and the integrated signature of microbial species, QTA, and other clinicopathological parameters.

The correlation of bacterial taxa with ICI efficacy was reported by multiple studies in recent years [20,21,22,23], possibly as part of an intricate crosstalk through the gut–lung axis [37,38]. Here, we show that Bacteroides dorei and Parabacteroides distasonis were associated with long OS in advanced-stage NSCLC patients. The increased abundance of Bacteroides species, including B. dorei, have been implicated in inflammation in several gut diseases, such as ulcerative colitis, irritable bowel disease [39], and celiac disease [40]. In contrast, B. dorei was also reported to reduce LPS production and inhibit atherosclerosis in a cohort of patients with coronary disease [41]. Interestingly, the same species was shown to increase ICI efficacy in colorectal cancer [42] but was associated with decreased survival time in melanoma [4]. A microbiome study on a multi-cancer patient cohort revealed that phylogenetically related species to B. dorei, B. xylanisolvens, and B. ovatus were significantly enriched in ICI responders and showed a synergistic effect with a combination erlotinib treatment [43]. A possible explanation for this contradiction might be the difference in cancer phenotype or the disparate average age of the patients at the time of diagnosis, since melanoma patients represent a relatively younger population. According to the latest research, P. distasonis is implicated in the pathogenesis of obesity and metabolic dysfunctions via bile acid and succinate production [44], but no association has been revealed so far with malignant diseases.

In contrast, *C. perfringens*, *E. faecium*, and *C. celatum* showed increased abundances in short OS patients. Clostridium species were reported to promote the accumulation of CD4+ T regulatory cells (Tregs) in colonic mucosa and were associated with Treg cell accumulation in colon cancer (CRC) [45,46]. Interestingly, while Treg cells were reported as positive prognostic factors in CRC, they are negative prognosticators in lung cancer [8], which may explain the diverse role of the taxon.

A unique feature of our study is the implementation of the mycobiome in ICI-related MG analysis, where we revealed a number of fungal taxa associated with OS and other clinical parameters. Cortinarius davemallochii, Helotiales, Chaetosphaeriales, Tremellomycetes and other fungal taxa were more abundant in patients with short OS. Also, in comparison, more fungal than bacterial taxa were associated with short OS, indicating that intestinal over-colonization of fungi might lead to decreased ICI efficacy.

In our study, antibiotic usage (either before or after ICI) did not contribute to a detriment in the ICI response or OS. This is in contrast with the studies of Derosa et al. [47] and Pinato et al. [48], but in line with other studies, where multivariate analysis could not underpin the significant negative effect of antibiotic usage on response or OS [23,49,50,51]. The toxicity of ICI was associated with the fungal taxon Thelephoraceae [52,53] and the bacterial genus Lachnospiraceae. The latter was reported to produce a considerable amount of short-chain fatty acids and to be associated with diabetes, metabolic syndrome [54,55], and inflammatory bowel disease [56]. When analyzing PD-L1 expression, we found associations with Dorea formicigenerans and Lachnospiraceae bacterium among other taxa correlating negatively with PD-L1. In contrast, other species, including the fungal taxa Hymenoscyphus immutabilis, Clavulinopsis fusiformis were associated with high PD-L1 expression.

Multiple research groups thoroughly studied the pulmonary mycobiome and identified Saccharomyces and Aspergillus fungi associated with a higher frequency of exacerbations and mortality in COPD patients [57,58]. In contrast, little is known about the gut mycobiome in COPD patients. In our study, the value of FEV1 showed a significant negative correlation with Clostridium celatum and multiple fungal species, including Cortinarius davemallochii, Thelephoraceae and Helotiales. Clostridium celatum is a Gram-positive anaerobic bacterium whose intestinal over-colonization with other Clostridium species frequently occurs after the antibiotic use that is broadly administered during COPD’s acute exacerbations [59]. The rudimental effects of PPI and H-blocker administration on the gut microbiome have been extensively studied [60,61]. However, while most studies focus on bacteria, the enrichment of fungal species, such as Serendipitaceae and Hyphodiscus, is a novel and intriguing finding in the field that might have future clinical implications.

From the individual QTA parameters, the value of Coarseness was significantly increased in the long OS and PD-L1-high patients, while the values of Energy and Kurtosis were significantly increased in the short OS and PD-L1-low patients, which may provide a rationale for selecting patients for immunotherapy. The multiple variables subtracted from QTA can be compressed to more manageable data components through PCA. PCs represent CT-derived key parameters that store valuable information that can be cost-effectively integrated into the clinical practice. In our study, seven PCs were identified, assembled from variations of the 105 measured QTA parameters. PC5 showed the highest number of associations with other parameters, apart from gender and COPD; PC5 was also associated with pseudoprogression. PC7 was significantly associated with the prediction of the response to ICI therapy. To underpin the role of QTA in predicting response, OS, and other clinical parameters, correlation studies in larger numbers of patient cohorts and in vivo validation in animal models, such as fecal microbiota transplantation (FMT), are required in the future [62].

To generate predictions for survival, response to ICI therapy, and other clinicopathological parameters, we used a novel machine learning approach, along with MG and QTA signatures. A peculiar advantage of the XGB algorithm is that, unlike all the other learning algorithms, XGB can account for missing data to enhance prediction. As the datasets usually include missing values, this innovative methodology increased the proportion of data included in the analysis. Using all the available parameters, the XGB machine learning algorithm showed therapy response associations with an accuracy of 83%, and distinguished long-term survival from short-term survival patients with an accuracy of 69%. While the filled XGB algorithm worked with peculiarly high PPVs for parameters like toxicity, PD-L1 expression (≥1%), response to therapy, and OS, it worked with relatively low NPVs for both ICI response and OS.

A chest CT scan is a routine examination at the time of NSCLC diagnosis. Therefore, it is a widely accessible data type that can, by using ML models, add predictive value to routine biomarker evaluation, such as PD-L1 expression, the tumor mutation burden (TMB), or the Immunoscore, in predicting ICI response. Despite being biologically intriguing, QTA features alone do not represent reliable predictive power in the clinical setting to assess ICI response or OS as they have accuracies of only 69% and 72%, respectively, based on the ML model. However, together with the Metagenome and other clinical parameters, a much more accurate predictive and prognostic nomogram could be established, with accuracies of up to 87% and 83%, respectively, further strengthening the narrative that assessing isolated biomarkers alone is not as effective as combining the predictive power of multiple biomarkers, using advanced algorithms.

The limitations of this study include the relatively low number of patients and the amount of available data. Furthermore, we did not have information on the immune microenvironment of the tumor samples, only the PD-L1 expression. Also, the patients were not assessed separately with regard to whether they received first-line ICI or a subsequent line, but the line of therapy did not show a correlation or significant association with response, OS, or a distinct metagenomic signature. The algorithm, however, was able to validate its accuracy through a training and prediction dataset. Thus, the present study is mainly for the generation of a hypothesis that needs further validation.

## 5. Conclusions

Our study reports significant associations of ICI efficacy and specific gut microbial species, QTA features, and clinicopathological parameters. Our data suggest that an integrated signature of the variables might predict outcomes with higher accuracy compared to the individual parameters in possible future therapeutic applications. Evaluating the machine learning algorithms based on contrast CT images and naïve Bayes and k-means clustering may predict the outcomes of immunotherapy. Further studies on “Big Data” are needed to define the exact prognostic value of QTA in NSCLC patients receiving ICI therapy.

## Figures and Tables

**Figure 1 cancers-15-05091-f001:**
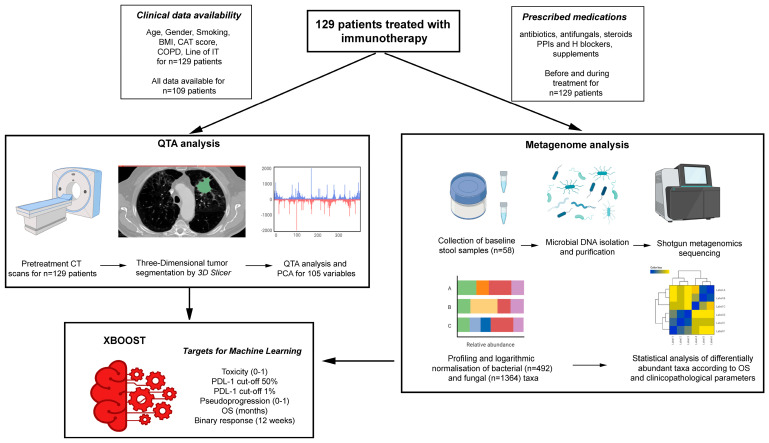
Study design and workflow, characteristics of patient cohorts, and availability of clinical data.

**Figure 2 cancers-15-05091-f002:**
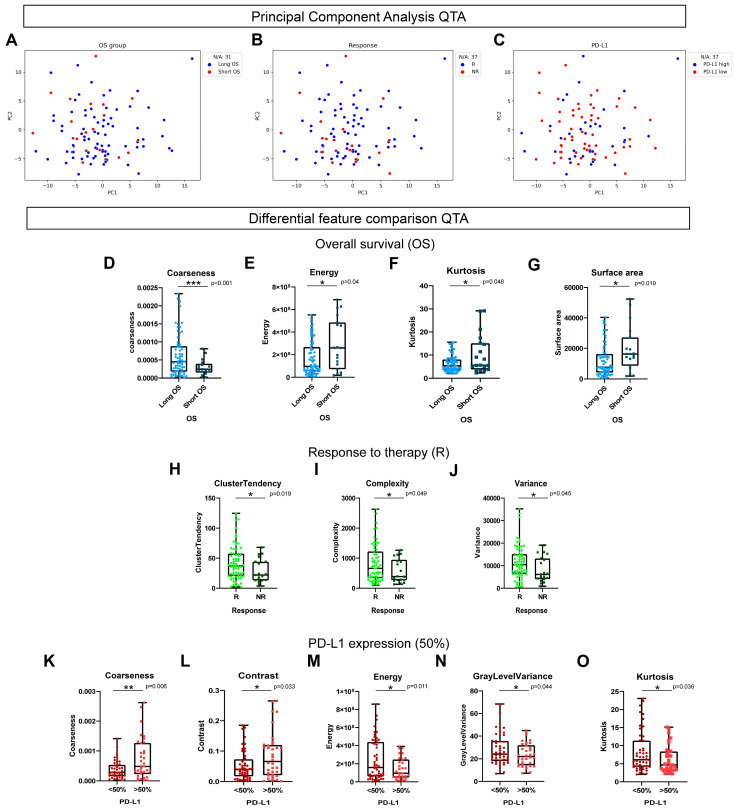
Results of principal component analysis (PCA) and individual QTA parameters showing significant differences according to OS, therapeutic response, and PD-L1 expression. Scatter plots show distribution of patient groups according to PC1 and PC2 derived from their QTA feature composition with PCA (**A**–**C**). PC1 and PC2 together explain 60.4% of variance in the QTA data matrix. Patients with missing clinical data were excluded from scatter plots (N/A). Bar charts show values of key QTA parameters in different patient groups. Coarseness was increased in long OS compared to short OS patients (0.0006288 vs. 0.0002974, *p* = 0.0004 (**D**), Energy (288,073,336 vs. 166,879,425, *p* = 0.0401 (**E**), Kurtosis (10.41 vs. 6.433, *p* = 0.048 (**F**), and surface area (19,317 vs. 12,086, *p* = 0.0190 (**G**) were increased in short OS compared to long OS patients. In responder patients, the value of QTA parameters ClusterTendency (42.83 vs. 28.82, *p* = 0.0194 (**H**), Complexity (822.3 vs. 579.7, *p* = 0.049, (**I**), and Variance (11379 vs. 8152, *p* = 0.0453 (**J**) were significantly increased (vs. non-responders). In PD-L1-high (≥50%) patients, Coarseness (0.0007438 vs. 0.0003818, *p* = 0.0069 (**K**) and Contrast (0.08508 vs. 0.05354, *p* = 0,0335 (**L**) showed increased values, whereas Energy (143,577,510 vs. 241,900,299, *p* = 0.0111 (**M**), GrayLevelVariance (23.44 vs. 28.52, *p* = 0.0442, (**N**), and Kurtosis (6.232 vs. 8.222, *p* = 0.038 (**O**) showed decreased values compared to PD-L1-low patients. Mann–Whitney U test was used to compare individual QTA parameters in different patient groups. Metric data were shown as median and corresponding standard deviation (SD), and graphs indicate the corresponding 95% CI. Statistical significance * *p* < 0.05; ** *p* < 0.01, *** *p* < 0.001.

**Figure 3 cancers-15-05091-f003:**
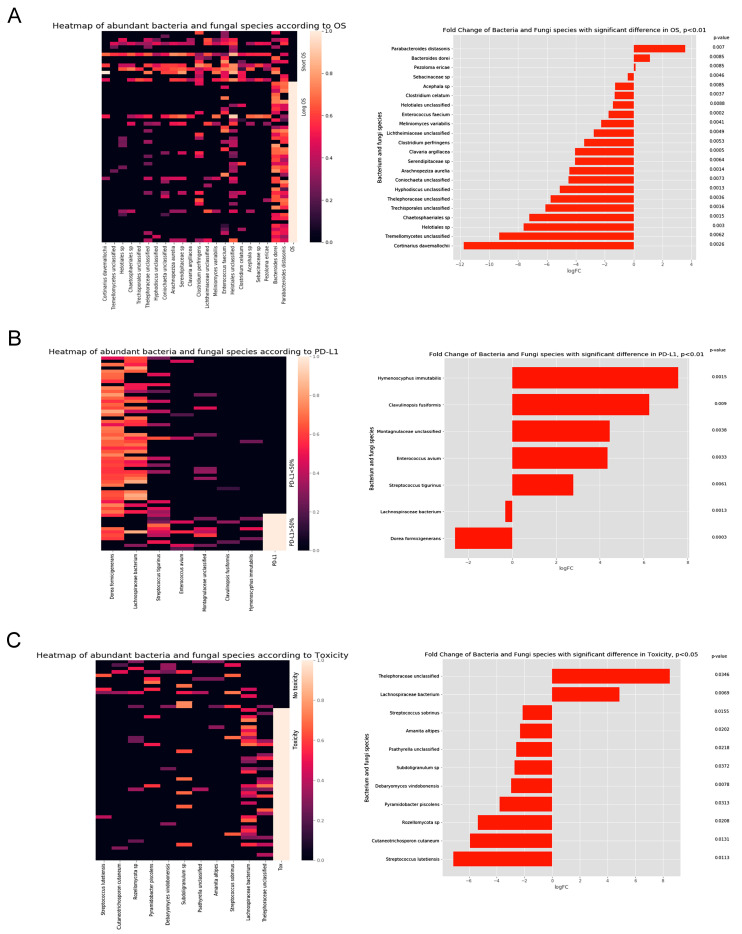
Abundant bacterial and fungal species according to overall survival (OS), PD-L1 expression, and toxicity. Heatmaps and bar plots represent abundant bacterial and fungal taxa enriched in patients according to OS (**A**), PD-L1 expression (<50% vs. ≥50%) (**B**), and toxicity (**C**). A total of 492 bacteria and 1364 fungi species were analyzed. From these, 901 species were present in both the long OS and the short OS groups. Heatmaps demonstrate taxonomic units with a significantly different abundance according to OS, PD-L1 expression, or toxicity. The range of the expressions was from 0 to 100. In bar plots, the *X* axis represents log fold change values for microbial taxa associated with different parameters. The *Y* axis represents *p*-values.

**Figure 4 cancers-15-05091-f004:**
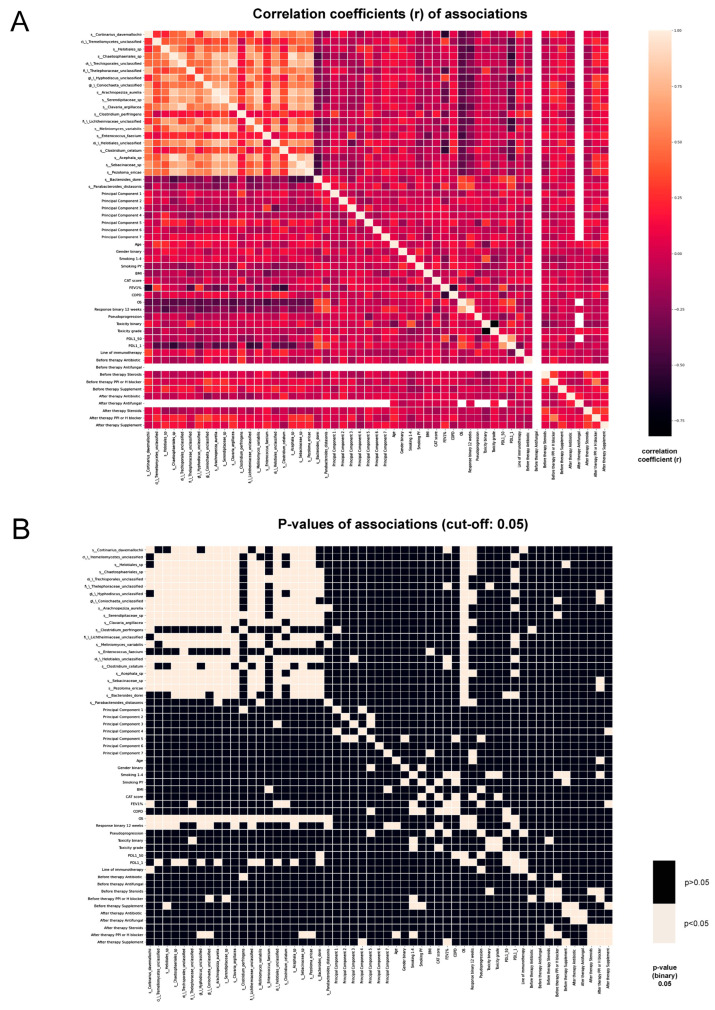
Correlation of abundant bacterial and fungal species, PCAs, clinicopathological parameters, prescribed other drugs, and response to therapy. Heatmaps show the distribution of correlation coefficients according to Spearman’s rank correlation (**A**) and the presence of statistical significance in a binary fashion (**B**), *p*-value < 0.05.

**Figure 5 cancers-15-05091-f005:**
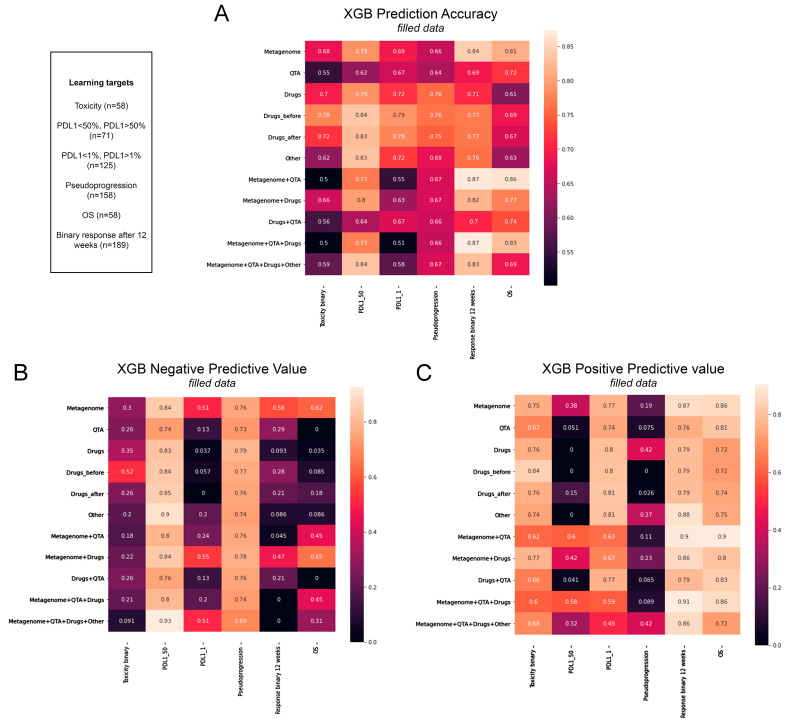
The accuracy, PPV, and NPV of the XGB machine learning algorithm used in predicting outcomes according to clinicopathological signatures of patients. The label “Other” (*Y* axis, **A**–**C**) means other key clinical parameters, including age, gender, smoking, PY, CAT score, FEV1%, COPD, line of ICI. The label “Drugs” (*Y* axis, **A**–**C**) means all kinds of administered drugs pooled together before or during therapy. Fractions in colored cells of matrices represent accuracy, NPVs, or PPVs (**A**–**C**). *PPV (positive predictive value), NPV (negative predictive value)*.

**Table 1 cancers-15-05091-t001:** Clinical parameters of patients according to OS group. *p*-values represent comparison between long OS and short OS patient groups, where a chi-squared test was used in the case of categorical variables and the Mann–Whitney U test was used in the case of continuous variables. *Toxicity = occurrence of any kind of irAE during ICI-treatment. CR = complete response; PR = partial response; SD = stable disease; PD = progressive disease. Statistical significance ** p < 0.01, *** p < 0.001*.

Clinical Parameter	Long OS Patients	Short OS Patients	*p*-Value
gender			0.802
male	41%	45%	
female	59%	55%	
age (years, mean)	65.07	61.5	0.086
PD-L1 expression			0.798
TPS ≥ 50%	41.1%	36.8%	
TPS < 50%	58.9%	63.2%	
Smoking (PY, mean)	39.45	37.53	0.658
BMI (kg/m^2^, mean)	25.95	24.67	0.395
COPD-comorbidity			>0.999
yes	32.7%	30.8%	
no	67.3%	69.2%	
CAT score (mean)	10.96	19.07	0.0056 **
FEV1% (mean)	69.93	64.7	0.54
Pseudoprogression			0.718
yes	18.1%	21.4%	
no	81.9%	78.6%	
Toxicity			>0.999
yes	65.7%	80%	
no	34.3%	20%	
Toxicity grade (mean)	1.06	0.778	0.157
Line of IT (mean)	1.85	1.5	0.088
Line of IT binary			0.123
first line	29%	47%	
subsequent line	71%	53%	
ICI Response at 3 months			<0.001 ***
Response (CR, PR, SD)	92%	26%	
Non-response (PD)	8%	74%	

## Data Availability

Processed data presented in this study are available as Appendix A. Raw sequencing data are available on reasonable request from the corresponding author.

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
