# Peer review of "Computed Tomography-Based Quantitative Texture Analysis and Gut Microbial Community Signatures Predict Survival in Non-Small Cell Lung Cancer"

_cancers, 2023, doi:10.3390/cancers15205091_

Round 1

Reviewer 1 Report

Dr. David Dora and colleagues on the impact of the relationship between CT-based QTA and the gut microbiome on the prognosis of NSCLC. However, in this paper, several assumptions were not considered in assessing the efficacy of immune checkpoint inhibitors.

My comments are listed below.

Major comments:

1.        Efficacy assessment should be conducted separately for patients undergoing first-line therapy and those undergoing second-line therapy. This differentiation is necessary due to the anticipated variability in the efficacy of ICIs for each group, compounded by the presence of patient selection bias.

2.        The efficacy of ICI cannot be assessed by referring only to OS, especially in retrospective studies, because post-treatment and prognostic factors influence OS. Particularly in retrospective studies, given the significant impact of post-treatment factors and prognostic variables on OS outcomes. Including data on PFS and ORR could provide additional support for the authors' viewpoint.

3.        Considering comments 1 and 2, it is advisable for the authors to thoroughly analyze whether QTA genuinely holds predictive value for ICI efficacy, accounting for variations in patient characteristics and treatment. As of the current information available, I cannot establish a conclusive link between CT-based QTA and gut microbiome concerning the prognosis of ICI.

Author Response

Major comments:

  1.       Efficacy assessment should be conducted separately for patients undergoing first-line therapy and those undergoing second-line therapy. This differentiation is necessary due to the anticipated variability in the efficacy of ICIs for each group, compounded by the presence of patient selection bias.

Response:

We thank the Reviewer for raising the point of efficacy. We collected samples prospectively; however, the data analysis was performed retrospectively and included first-line (n=35) and subsequent-line (n=62) cases. Our data differed from prospective clinical trials and resulted in only slight differences in efficacy across lines of therapy. We analyzed the line of therapy and incorporated it into Table 1. There were no significant differences in long vs short OS cases according to the line of treatment in this “real world” dataset. According to the Reviewers' suggestions, we incorporated response and also, there were no significant differences in response rate according to the line of therapy (first- vs subsequent line, chi-squared test, p=0.097) or long vs short OS groups (chi-squared test, p=0.123). Also, we included this in the discussion as a limitation, adding that there were no significant differences in microbiome composition according to the line of therapy. Interestingly, in our previous study, we drew the same conclusion on a different immunotherapy (IT)-treated NSCLC cohort, when assessing progression-free survival (cut-off: 6 months) (Dora et al, 2023). Briefly, chemotherapy (before the first IT cycle) alters the gut microbiome, but not in a way that would significantly affect IT efficacy. Removed sections appear with strikethrough and track change, and newly added sections appear with green highlighting in the manuscript.

Dora D, Ligeti B, Kovacs T, Revisnyei P, Galffy G, Dulka E, Krizsán D, Kalcsevszki R, Megyesfalvi Z, Dome B, Weiss GJ, Lohinai Z. Non-small cell lung cancer patients treated with Anti-PD1 immunotherapy show distinct microbial signatures and metabolic pathways according to progression-free survival and PD-L1 status. Oncoimmunology. 2023 May 12;12(1):2204746. doi: 10.1080/2162402X.2023.2204746. PMID: 37197440; PMCID: PMC10184596.

  1.       The efficacy of ICI cannot be assessed by referring only to OS, especially in retrospective studies, because post-treatment and prognostic factors influence OS. Particularly in retrospective studies, given the significant impact of post-treatment factors and prognostic variables on OS outcomes. Including data on PFS and ORR could provide additional support for the authors' viewpoint.

Response:

We thank the Reviewer for pointing out that PFS is considered an indication of disease control and stabilization. However, in this specific study, we did not use PFS as an outcome and learning target in the machine learning model, instead used a shorter and a longer-term variable to represent the outcome-spectrum: Response and OS. Therefore, we included response defined as the percentage of patients with stable disease or who achieve a response to therapy at 12 weeks after immunotherapy initiation. We used instead PFS as our primary outcome in our previous article referenced above (Dora et al, 2013)

  1.       Considering comments 1 and 2, it is advisable for the authors to thoroughly analyze whether QTA genuinely holds predictive value for ICI efficacy, accounting for variations in patient characteristics and treatment. As of the current information available, I cannot establish a conclusive link between CT-based QTA and gut microbiome concerning the prognosis of ICI.

Response:

We thank the Reviewer for pointing out the importance of clinicopathological parameters. Therefore, we included as much as possible in our data collection and analysis. Among others, we had prescribed medications with potential effects on microbiome composition or outcomes, such as antibiotics, antifungals, steroids, PPIs, H blockers, and supplements before or during treatments separately. Additionally, we analyzed the associations of clinicopathological characteristics such as age, gender, PD-L1, smoking pack year, COPD, and lung function parameters (FEV1). We identified no significant differences according to long vs short OS. However, there were increased COPD assessment test scores (CAT) in short OS cases (Table 1). Although we found no substantial effects of the individual parameters, by incorporating them into the model, we hoped to identify signatures to predict survival better than the traditionally used individual parameters. 

In our manuscript, we highlighted that QTA alone is not a reliable predictive of prognostic factor, only when it is supplemented with other clinicopathological parameters and metagenomic signatures (lines 555-562). The rationale behind including specifically these two seemingly distinct parameters (QTA and metagenome) can be explained by the non-invasive, cost-effective, and widely accessible nature of CT scans and the innovative emerging nature of fecal metagenome-analysis and machine learning algorithms in the field of oncology that currently progresses with an enormous rate and potentially be incorporated in routine patient diagnostic in a couple of years.

Accordingly, our data suggests that an integrated signature of the variables might predict outcomes with higher accuracy compared to the individual parameters with possible future therapeutic applications.  

Reviewer 2 Report

I have the following comments on this interesting manuscript, which describes an overall well designed study:

-The Abstract is rather long and should be reduced by about 30%. The results of the PCA (lines 34-36) should be briefly illustrated also in the main text. Also the Discussion is very long and could be reduced by about 30-50% by focusing as much as possible on the interpretation of the study findings and their comparison with the existing literature.

-Section 2.3. Who performed 3D tumor segmentation? How many readers, and with which kind and degree of experience? How were discordant findings dealt with (consensus reading? other?)

-Section 2.10 ('XGBoost models for classification') is way too long and should be substantially reduced by leaving or adding references to the relevant literature.

The English language needs some minor improvements.

Author Response

I have the following comments on this interesting manuscript, which describes an overall well designed study:

-The Abstract is rather long and should be reduced by about 30%. The results of the PCA (lines 34-36) should be briefly illustrated also in the main text. Also the Discussion is very long and could be reduced by about 30-50% by focusing as much as possible on the interpretation of the study findings and their comparison with the existing literature.

Response:

We thank the Reviewer for the positive attitude towards our manuscript and for pointing out these important issues. First, we included the illustration of the PCA results in Figure 1 (A-C), showing the first 2 PCs on the plot that explained roughly 60% of the QTA data variance. Patients with no available data regarding the 3 observed parameters (OS, Response, PD-L1 expression) were excluded. We also added the explanation in the Figure Legend (lines 313-317) and in the main text lines 297-299. 

Based on the suggestions of the expert Reviewer, we also shortened the Discussion significantly, instead adding more about the scientific rationale of the study (main text, lines 553-560). Removed sections appear with strikethrough and track change, and newly added sections appear with green highlighting in the manuscript.

-Section 2.3. Who performed 3D tumor segmentation? How many readers, and with which kind and degree of experience? How were discordant findings dealt with (consensus reading? other?)

Response:

We thank the Reviewer for pointing out the issue of standardization. Based on a standardized radiology report that was part of standard-of-care performed using the RECIST 1.1 criteria by an experienced radiologist. Next, a board-certified radiologist reevaluated the tumor region of interest. We defined the primary tumor as the largest lesion in diameter surrounded by lung parenchyma. The lesion should not be centrally located, directly infiltrating, or connected with other tissue compartments, the mediastinal area, or the chest wall.

The 3D segmentation was performed using an automated (Fast GrowCut) and robust algorithm to segment the volume fully. We Used Label 1 as the foreground and Label 2 as the background. Multiple label colors can define regions representing parts of anatomical structures. The standardized algorithm will seek the best labeling for an adjacent pixel to match the lesion. Therefore, there was no need for consensus reading using the automated segmentation process. Accordingly, we added a paragraph on this in the methods section (lines 144-158). 

-Section 2.10 ('XGBoost models for classification') is way too long and should be substantially reduced by leaving or adding references to the relevant literature.

Response:

As requested we removed parts from the corresponding methods section that are not essential for the understanding of the ML model and documented well on the webpage of the XBoost algorithm. However, it was not possible to add relevant literature XBoost is a relatively novel ML algorithm, thus we have not found any studies using it for biomedical research. However, we provided the hyperlinks for detailed documentation.

Reviewer 3 Report

This paper present using CT QTA and biomaker signature to predict overall survival of ICI-treated NSCLC patients, and they also used the XGB model to evaluate associations between outcomes and various clinicopathological parameters. Overall, this study is very interested, and would help the research in this field. However, there are some issues that required the authors to improve the manuscript.

1. For the data pre-processing, the authors mentioned they used SKlearn deep learning method. The authors may provide detailed operations on how they processed the CT scans. For PCA, how did the authors extract the key components? Using each CT scan as the PCA input? Since the data usage of PCA is kind of low and predict efficiency is also low when compared to deep learning methods. Do the author consider using the deep learning method to extract the features from CT scans? If so, the author may combine PCA, DL, and  ICI together.

2. For words in Line 267 and 268, the authors mentioned 129 patients were enrolled in this study, but Table 1 only indicate the male data, how about the female information? Were they excluded from the study? 

3. In Fig.1, the authors mentioned 109 paitents, what are the 109 patients standing for? Also, the authors may also provide some informations about the XBOOST model.

4. Fig. 2k, the X-label is missing.

Author Response

This paper present using CT QTA and biomaker signature to predict overall survival of ICI-treated NSCLC patients, and they also used the XGB model to evaluate associations between outcomes and various clinicopathological parameters. Overall, this study is very interested, and would help the research in this field. However, there are some issues that required the authors to improve the manuscript.

  1. For the data pre-processing, the authors mentioned they used SKlearn deep learning method. The authors may provide detailed operations on how they processed the CT scans. For PCA, how did the authors extract the key components? Using each CT scan as the PCA input? Since the data usage of PCA is kind of low and predict efficiency is also low when compared to deep learning methods. Do the author consider using the deep learning method to extract the features from CT scans? If so, the author may combine PCA, DL, and  ICI together.

Response:

We thank the Reviewer for bringing this issue up. Three-dimensional tumor segmentation was performed using the 4.10 version of 3D Slicer, and a total of 105 CT parameters from each CT image were obtained. The 3D segmentation was performed using an automated (Fast GrowCut) that is an in-built robust algorithm to segment the volume fully. We Used Label 1 as the foreground (tumor) and Label 2 as the background (lung parenchyma). Multiple label colors can be used to define regions representing parts of anatomical structures. The algorithm will find the best labeling for an adjacent pixel to match the tumor volume. 

Next, we used the Pyradiomics software package (https://pyradiomics.readthedocs.io/en/latest/features.html) to obtain the QTA parameters from the segmentation masks and the CT scans. This is roughly 105 numerical values, some of which are easy to interpret, e.g. the mean, range, and entropy of the Hounsfield intensities, and others that are more difficult, e.g. the complexity of the Neighbouring Gray Tone Difference Matrix. PCA selects the vectors of the original feature space that best explain the discrepancies. E.g. in Pyradiomics there are separate features for voxelvolume and meshvolume, both contain the size of the segmented tumor, but are calculated differently and may differ by a few percent. Feature compression (dimensionality reduction) is an important step, because in a small number of cases when a model is finally fitted, each additional input feature increases the chance of overfitting and reduces robustness. 

In summary, the input to PCA was QTA feature vectors computed on CTs with lower information content (roughly the 100 dimensions mentioned above). Although the possibility was explored, the number of images included in the study did not allow the development of feature extraction using our own CT-based deep learning (this would require at least 1000 CTs, if not more), and the study participants were not aware that there was/is such a model trained by others that could be used for prediction. The method we present and the CT feature extraction obtained by Pyradiomics are a realistic alternative to deep learning algorithms for statistically small-scale research studies like the current ones, which are more robust and less resource-intensive. Accordingly, we added more detailed documentation in the Methods section, lines 144-158. Removed sections appear with strikethrough and track change, and newly added sections appear with green highlighting in the manuscript.

  1. For words in Line 267 and 268, the authors mentioned 129 patients were enrolled in this study, but Table 1 only indicate the male data, how about the female information? Were they excluded from the study?

Response:

We apologize for not indicating clearly the characteristics in Table 1. Percentages in the columns “Long OS vs Short OS applies to categoricals in brackets if present (e.g.: male) or to the presence of the characteristics (e.g.: COPD=yes) if not indicated. We edited Table 1, to be more informative and clear and now both the male and female percentages in the Long OS and Short OS groups are present, just like in the case of other parameters.

  1. In Fig.1, the authors mentioned 109 paitents, what are the 109 patients standing for? Also, the authors may also provide some informations about the XBOOST model.

Response:

In Fig 1. it is indicated that “all data available for n=109 patients”, meaning that there are 109 patients in the whole cohort (n=129), who had all clinicopathological variables documented, including gender, age, OS, Response to therapy, PD-L1 expression, Smoking, BMI, COPD-comorbidity, Toxicity, Toxicity-grade, CAT score, FEV1%,  Pseudoprogression and the line of IT. All 129 patients had QTA features documented, and n=58 patients had their fecal samples undergone metagenome and ITS sequencing. Detailed description of the XBoost model used are shown in the Methods section, lines 247-283.

  1. Fig. 2k, the X-label is missing.

Response:

We thank the Reviewer for spotting this error, we added the missing annotation.

Reviewer 4 Report

i have some feedback regarding three areas that could benefit from improvement. Primarily, the synopsis consists of an abundance of particulars that do not add to the principal communication. A shorter and more concise abstract would allow readers to better understand the study's main objectives and points. An AI is unlikely to produce a sentence that describes a study project's goal in predicting the survival rate of non-small cell lung cancer patients treated with immune checkpoint inhibitors through a combination of microbiome-based biomarker signatures and computed tomography-based texture analysis. The information was amassed from CT scans and fecal microbiome samples of 129 NSCLC patients who underwent anti-PD-L1 immunotherapy. QTA parameters and microbial species were identified and associated with OS, response to therapy, PD-L1 expression, and toxicity using an artificial intelligence (AI) approach based on Extreme Gradient Boosting. The results show that an integrated signature of these characteristics may predict outcomes more accurately than separate measures.

Furthermore, the opening exhibits an absence of a distinct research query and proposed assumption. The authors are required to distinctly declare their research goals and anticipated consequences. This would assist readers in navigating the background information and rationale of the study. According to our hypothesis, QTA parameters and gut microbiome signatures can predict OS, response to therapy, PD-L1 expression, and toxicity in NSCLC patients treated with ICI, and a machine learning approach can combine these variables to create a reliable predictive model, as we suggest in this research.

Finally, the results section is not well-organized and lacks a logical order. The authors must present their findings in a lucid, coherent manner, maintaining the same sequence as the approaches section. They should also use appropriate subheadings to separate different parts of the results section. For example, the results section could be divided into four subheadings: QTA parameters can predict OS, response to therapy and PD-L1 expression; Microbial taxonomic profiling reveals associations with OS, response to therapy, PD-L1 expression, and toxicity; Outcomes predicted by the XGB machine learning algorithm; and Validation of the predictive model in an independent cohort.

The manuscript employs lucid and succinct diction, with fitting technical terminology and definitions. The manuscript adheres to a coherent framework and furnishes ample details and explications for the procedures and outcomes.

The manuscript exhibits certain grammatical and orthographical inaccuracies, for instance, "mineable data" (line 62), "tu-mor-infiltrating immune cells" (line 65), "tu-mor microenvironment" (line 66), "non-invasively assessing" (line 59), and "theranostic role" (line 531). The manuscript also manifests some incongruous punctuation and formatting, including omitted commas, periods, and spaces amid words or sentences.

Author Response

I have some feedback regarding three areas that could benefit from improvement. Primarily, the synopsis consists of an abundance of particulars that do not add to the principal communication. A shorter and more concise abstract would allow readers to better understand the study's main objectives and points. An AI is unlikely to produce a sentence that describes a study project's goal in predicting the survival rate of non-small cell lung cancer patients treated with immune checkpoint inhibitors through a combination of microbiome-based biomarker signatures and computed tomography-based texture analysis. The information was amassed from CT scans and fecal microbiome samples of 129 NSCLC patients who underwent anti-PD-L1 immunotherapy. QTA parameters and microbial species were identified and associated with OS, response to therapy, PD-L1 expression, and toxicity using an artificial intelligence (AI) approach based on Extreme Gradient Boosting. The results show that an integrated signature of these characteristics may predict outcomes more accurately than separate measures.

Furthermore, the opening exhibits an absence of a distinct research query and proposed assumption. The authors are required to distinctly declare their research goals and anticipated consequences. This would assist readers in navigating the background information and rationale of the study. According to our hypothesis, QTA parameters and gut microbiome signatures can predict OS, response to therapy, PD-L1 expression, and toxicity in NSCLC patients treated with ICI, and a machine learning approach can combine these variables to create a reliable predictive model, as we suggest in this research.

 Finally, the results section is not well-organized and lacks a logical order. The authors must present their findings in a lucid, coherent manner, maintaining the same sequence as the approaches section. They should also use appropriate subheadings to separate different parts of the results section. For example, the results section could be divided into four subheadings: QTA parameters can predict OS, response to therapy and PD-L1 expression; Microbial taxonomic profiling reveals associations with OS, response to therapy, PD-L1 expression, and toxicity; Outcomes predicted by the XGB machine learning algorithm; and Validation of the predictive model in an independent cohort.

Response:

We thank the Reviewer for their insightful comments and benign, constructive criticism. To emphasize the rationale of the study, and to elaborate the aim to show that a multi-approached, integrated signature could possess a stronger predictive and even prognostic power than analyzing these parameters alone. Therefore, we added 1-1 paragraph in the Abstract, (lines 48-51) and in the Introduction section (lines 95-97) to highlight this important rationale using the proposed text by the expert Reviewer. Moreover, we reorganized the Results section, removing some redundant information and using the proposed subheadings. Removed sections appear with strikethrough and track change, and newly added sections appear with green highlighting in the manuscript. 

Comments on the Quality of English Language

The manuscript employs lucid and succinct diction, with fitting technical terminology and definitions. The manuscript adheres to a coherent framework and furnishes ample details and explications for the procedures and outcomes.

The manuscript exhibits certain grammatical and orthographical inaccuracies, for instance, "mineable data" (line 62), "tu-mor-infiltrating immune cells" (line 65), "tu-mor microenvironment" (line 66), "non-invasively assessing" (line 59), and "theranostic role" (line 531). The manuscript also manifests some incongruous punctuation and formatting, including omitted commas, periods, and spaces amid words or sentences.

Response:

We thank the Reviewer for pointing these inaccuracies out and we corrected the debated parts accordingly throughout the manuscript. Furthermore, we ran one more round of English language checks to improve the manuscript in this field as well.

Round 2

Reviewer 1 Report

I would like to thank you for your courteous response. From a clinical perspective, I understand the usefulness of this study. I believe the authors responded appropriately to my review.